# Applying a GAN-based classifier to improve transcriptome-based prognostication in breast cancer

**Cristiano Guttà**[1]\*, **Christoph Morhard**[2], **Markus Rehm**[1,3]\*

1 Institute of Cell Biology and Immunology, University of Stuttgart, Stuttgart, Germany, 2 ProKanDo GmbH, Ludwigsburg, Germany, 3 Stuttgart Research Center Systems Biology, University of Stuttgart, Stuttgart, Germany

\* cristiano.gutta@izi.uni-stuttgart.de (CG); markus.morrison@izi.uni-stuttgart.de (MR)

**Data Availability Statement:** Transcriptome data (median Z-scores), overall survival, disease specific survival DSS and associated clinical records of the METABRIC cohort were downloaded from cbioportal.org. Locoregional and distant

## Abstract

Established prognostic tests based on limited numbers of transcripts can identify high-risk breast cancer patients, yet are approved only for individuals presenting with specific clinical features or disease characteristics. Deep learning algorithms could hold potential for stratifying patient cohorts based on full transcriptome data, yet the development of robust classifiers is hampered by the number of variables in omics datasets typically far exceeding the number of patients. To overcome this hurdle, we propose a classifier based on a data augmentation pipeline consisting of a Wasserstein generative adversarial network (GAN) with gradient penalty and an embedded auxiliary classifier to obtain a trained GAN discriminator (T-GAN-D). Applied to 1244 patients of the METABRIC breast cancer cohort, this classifier outperformed established breast cancer biomarkers in separating low- from high-risk patients (disease specific death, progression or relapse within 10 years from initial diagnosis). Importantly, the T-GAN-D also performed across independent, merged transcriptome datasets (METABRIC and TCGA-BRCA cohorts), and merging data improved overall patient stratification. In conclusion, the reiterative GAN-based training process allowed generating a robust classifier capable of stratifying low- vs high-risk patients based on full transcriptome data and across independent and heterogeneous breast cancer cohorts.

## Author summary

Despite the considerable advance in prevention, diagnosis and patient management, breast cancer still represents the second leading cause of cancer death among women. Identifying those patients at high risk of tumor relapse or progression is crucial for optimal patient management. Currently, prognosis mainly relies on the expression of individual biomarkers or on multi-gene expression signatures. However, these signatures are approved for use only in subsets of patients with specific clinical characteristics. In this study, we set out to study if more general patient prognosis can be derived from full transcriptome profiles by a deep learning strategy. Our framework, named T-GAN-D, makes use of deep learning algorithms originally conceived for image analysis (i.e. generative

recurrence information and Risk of Recurrence – Proliferation scores were retrieved from retrieved from Rueda et al. (DOI: 10.1038/s41586-019-1007-8) and Xia et al. respectively (DOI: 10.1038/s41467-019-13588-2). Clinical records, overall survival, disease specific survival and progression free interval of the TCGA-BRCA cohort were integrated from cbioportal.org and Liu et al. (DOI: 10.1016/j.cell.2018.02.052) respectively. Normalized METABRIC and BRCA expression data were downloaded using the R package MetaGxBreast (Gendoo et al., DOI: 10.1038/s41598-019-45165-4). On Zenodo (DOI: 10.5281/zenodo.7151831) are available the following data and scripts: (1) the R script used for downloading, rescaling and merging the METABRIC and MRCA-TCGA cohorts used as use cases in the study; (2) the input files (clinical and expression data) used to train and test the T-GAN-D as described in the article are provided; (3) the Python scripts used for data pre-processing and generation of risk class predictions.

**Funding:** MR and CG receive funding by the Deutsche Forschungsgemeinschaft (DFG, German Research Foundation) under Germany's Excellence Strategy - EXC 2075 – 390740016. CG received a salary from the aforementioned funder at the time of the study. The funders had no role in study design, data collection and analysis, decision to publish, or preparation of the manuscript.

**Competing interests:** I have read the journal's policy and the authors of this manuscript have the following competing interests: Christoph Morhard is CEO of ProKanDo, a company offering IT support and consultation, including the area of medical technology. CM declares no competing financial interests in relation to the research results presented in this study. The other authors likewise declare no competing interest.

adversarial networks) to discriminate high and low risk breast cancer patients. For this, gene expression profiles are converted into images, in which each pixel represents the expression of one transcript. These images are then used as input for a classifier that, after being trained, separates patients into the respective risk categories. Our results show that the T-GAN-D surpassed established prognostic biomarkers in performance.

## Introduction

Breast cancer is the tumor with the highest incidence in women, accounting for 2.3 million new diagnoses and 685,000 deaths worldwide in 2020. According to the World Health Organization, nearly eight million patients were diagnosed with breast cancer in the five years before 2020, making it the most prevalent tumor disease worldwide [1]. In current clinical practice, the expression of estrogen receptor (ER), progesterone receptor (PR), and human epidermal growth factor receptor 2 (HER2) is determined by immunohistochemistry (IHC), with the expression patterns defining to which molecular subtype (luminal A, luminal B, HER2 positive or enriched and triple-negative breast cancer) individual tumors belong. Prognosis differs between these subtypes, and subtyping informs treatment plans in patients in which surgical resection of the tumor alone is insufficient [2]. However, substantial response heterogeneities to the current standard of care treatments can be observed in populations of breast cancer patients [3], highlighting the need for additional prognostic markers that could serve to identify high risk patients that could instead benefit from alternative treatments or for which the burden from inefficient standard of care treatments could be avoided [4].

Various multi-gene activity tests based on transcript abundance have been developed to assist in the clinical management of breast cancer (e.g. Oncotype DX [5], MammaPrint [6,7], Prosigna [8,9], OncoMasTR [10]) and received regulatory approval as prognostic tests [11]. Despite the prognostic value of these assays, their use is restricted to only subsets of patients with specific clinical characteristics (e.g. cancer stage, receptor or lymph node status, tumor size, menopause state, age group) [12–14]. It would therefore be desirable if more generally applicable prognostic tests based on transcriptome data could be developed.

The rapid advances in high-throughput sequencing technologies make tumor transcriptome data from larger patient cohorts increasingly available. The accessibility of -omics databases and companion clinical information now also encourages the application of deep learning (DL) methods to the oncology field, with the aim of learning and extracting features within large scale data that are not readily accessible by classical statistical and pattern recognition approaches. It is hoped that from DL-based methods can be developed tools that can aid in further advancing cancer diagnosis, prognosis or predicting treatment efficacy in the future [15].

DL algorithms such as convolutional neural networks (CNN) were originally applied to image analysis but could be successfully repurposed to take non-image objects as input, such as RNA-seq data [16]. One of the major pitfalls when applying DL models to transcriptome datasets is the typical imbalance between the number of quantified mRNAs (high) and the number of patients (low), which can lead to overfitting when solving classification tasks [17]. In addition, low numbers of samples or patients that represent one category (e.g. good prognosis) come at the risk of capturing patterns that are not robust when applied to larger populations [18]. Feature selection (FS) strategies [19], under- and over-sampling [20] are three strategies that may help mitigating effects arising from imbalanced source data. An alternative strategy lies in novel data augmentation approaches, such as generative adversarial networks (GANs), by which source datasets can be enriched with artificially generated additional data.

GANs are typically applied to imaging data and are composed of two subnetworks, the generator and the discriminator. While the former produces synthetic images, the latter is challenged to discriminate fake vs. real images. Reiterating this process, the generator learns to produce images with features that can no longer be separated from the real images by the discriminator, with these generated images then enriching the source dataset [21]. In comparison to other generative models, GANs are currently preferred due to their computational speed and the quality of the generated images [22]. In addition, they exhibit a lower risk of overfitting classifiers and are less susceptible to the impact of non-pertinent image features (such as brightness) when enriching training data with synthetic images [23]. For example, GANs have been applied in the medical field to generate synthetic magnetic resonance, computed tomography or positron emission tomography images [24]. Aside from image-data, different GAN implementations were also successfully applied to transcriptome data for cancer diagnosis [25,26], staging [27] and subtyping [28].

The Molecular Taxonomy of Breast Cancer International Consortium (METABRIC, hereafter MB) [29] and The Cancer Genome Atlas—Breast Invasive Carcinoma (TCGA-BRCA, hereafter TCGA) [30] cohorts represent two of the largest and most exhaustively annotated breast cancer datasets, including, in addition to mRNA expression data, features such as patient demographics, cancer staging, receptor statuses, and follow-up information such as survival times. Despite not being directly interoperable due to different sequencing technologies, these datasets can serve as use cases to test new DL-based prognostication approaches.

In this study, we therefore set out to develop a prognostication framework that used the trained discriminator of a GAN architecture (T-GAN-D) as a standalone classifier. To this end, we first tested if unstructured data such as transcriptome profiles could be used as input for DL architectures after conversion to images. Afterwards, in a transfer learning fashion, the T-GAN-D was used independently to predict the risk category of MB breast cancer patients. Subsequently, the robustness of our framework was assessed integrating patient profiles from the independent TCGA cohort in the training set. The stratification performance was compared to classic ML algorithms, a CNN and breast cancer biomarkers commonly used in clinical practice. Finally, the transferability of our framework was tested by applying the T-GAN-D to the smaller and imbalanced TCGA cohort.

## Materials and methods

### Data integration

The METABRIC dataset (n = 1904 patients, m = 18543 transcripts) was used to develop the prototype network implementation. Transcriptome data (median Z-scores), overall survival (OS), disease specific survival (DSS) and associated clinical records were downloaded from cbioportal.org [31,32]. The dataset was integrated with locoregional and distant recurrence information retrieved from Rueda et al. [33] and *Risk of Recurrence—Proliferation* (ROR-P) scores reported by Xia et al [9]. Clinical records, OS, DSS and progression free interval (PFI) of the validation TCGA-BRCA cohort (n = 1101 patients, m = 20532 transcripts) were integrated from cbioportal.org [31,32] and Liu et al. 2018 [34], respectively. To merge the mRNA expression data of the two cohorts, normalized transcriptome datasets were downloaded using the R package MetaGxBreast [35]. The transcript amounts were rescaled as described by Gendoo et al. [35] so that the 2.5 percentile corresponds to -1 and the 97.5 percentile corresponds to +1. Subsequently, transcripts overlapping between the two cohorts and with quantitative information missing in not more than five patients were retained. The resulting "*post-merging*" MB + TCGA dataset contains the expression values of m = 14042 genes. The R script used to download and rescale the datasets is available in the Zenodo repository [36].

## Inclusion criteria and category definition

Both cohorts were filtered to exclude normal-like subtype samples [9,37,38] and patients for which less than 10 years of follow-up time from diagnosis were available. Low and high risk categories were defined according to published clinical records [8,9] as follows:

- high risk patients:

  - MB cohort: disease specific death, locoregional or distant recurrence event recorded before 10 years from initial diagnosis;

  - TCGA cohort: disease specific death, progression, local recurrence or distant metastases before 10 years from initial diagnosis.

- low risk patients: none of the above-mentioned events recorded before 10 years from initial diagnosis.

A follow-up time of 10 years is sufficient to capture both early and late relapse events [39]. In addition, the chosen clinical endpoints allow a direct comparison of the classifiers with the ROR-P categories of the MB cohort published by Xia et al [9].

In total, 1248 patients of the MB cohort (n = 567 high risk, n = 681 low risk) and 165 patients of the TCGA cohort (n = 132 high risk, n = 33 low risk) satisfied the inclusion criteria. Four patients from each cohort were excluded after merging the two datasets due to insufficient expression data.

## GAN architecture

The architecture was based on a Wasserstein [40] GAN [21] with gradient penalty [41] and an auxiliary classifier [42] as a variant of a conditional GAN implementation [43], yielding a AC-WGAN-GP architecture. The Wasserstein loss was implemented to reduce vanishing gradients and mode collapse [44] in the early phases of the training when the discriminator outperformed the generator. Stability was improved by exchanging the weights clipping approach described in Arjovsky et al. [40], with the gradient penalty described in Gulrajani et al. [41]. To create a conditional GAN, an auxiliary classifier network was implemented [42], resulting in a more stable training process and reduced mode collapse compared to the standard conditional GAN approach, supplying labels to both discriminator and generator [44]. A z-vector of size 250 was fed as input for the generator. Following good training practice [45], strided convolutions with step size 2, batch normalization and LeakyRELU as activation function were used. Since using batch normalization in the discriminator and/or the ADAM optimizer led to an unstable training process, batch normalization [46] was only used in the generator, and RMSprop was selected as the activation function. A shallow network consisting of two layers in both the discriminator and the generator led to the most stable training process, due to the smaller number of trainable parameters compared to deeper networks. Hyperparameters were tuned empirically, selecting 1000 epochs for the training process. Three "discriminator-only" training runs were performed before each full network training run, and the generated pictures were subsequently smoothed with a final convolution layer with one filter and stride size of 1. The GAN architecture generated expression profiles of size 144 ×144 when using the MB dataset before merging the cohorts (m = 18453 transcripts), starting from a square matrix of size $18 \times 18$. After merging the MB and TCGA cohorts (m = 14042 overlapping transcripts), a starting matrix of size $15 \times 15$ was used, generating expression profiles of size 120 ×120. In both implementations, the values missing to populate the final square matrix were filled with random values sampled from a -1 to 1 uniform distribution. The resulting trained GAN

Discriminator (T-GAN-D) was then used as an independent classifier to discriminate low and high risk patients. The Python code and the input files used to generate the predictions are available in the Zenodo repository [36].

## CNN architecture

As the performance of the CNN implemented as the GAN's discriminator showed satisfactory performance, a similar architecture was used as a benchmark classifier. Batch normalization was employed to ensure shorter training periods and RELU was used as the activation function. A fixed training length of 1250 epochs was set due to the limited sample size and to generate comparable iterations.

## Feature selection and application of classic machine learning (ML) algorithms

Random Forest (RF) and Support Vector Machines (SVM) classifiers were implemented in the WEKA Workbench (Version 3.8) [47] using the classifiers *trees.RandomForest* and *functions. LibSVM* with default settings. Both algorithms were trained on the merged MB + TCGA cohort to predict the risk class of MB patients. A 5-fold cross validation was applied using the same training and test set splits to ensure comparability between the different methods. At each iteration, the attribute selection *SymmetricalUncertAttributeSetEval* with *FCBFSearch* (Fast Correlation-Based Filter [48]) method was applied to the training set. This preprocessing step aimed at selecting an optimal subset of transcripts with the highest correlation with the class and the lowest redundancy with other relevant features.

## Survival analysis and accuracy

Log-rank testing was used to compare predicted low vs high risk patients over a follow-up time of 10 years. Kaplan-Meier (KM) survival curves were computed using GraphPad Prism 8 (GraphPad Software, San Diego, CA). The area between the curves (ABC) displayed on the KM graphs for the pooled predictions was calculated as follows:

- Low risk AUC minus Predicted low risk AUC;

- Predicted low risk AUC minus Predicted high risk AUC;

- Predicted high risk AUC minus High risk AUC.

The ABCs values are shown on the graphs in the abovementioned order top to bottom. The AUC was computed using GraphPad Prism 8 (GraphPad Software, San Diego, CA). Univariate and multivariate hazard ratios were calculated using the function *coxph* from the R's library *survival* (v. 3.4.0, https://www.r-project.org/).

The accuracy of all classifiers was calculated dividing the number of correct classifications by the total number of classifications performed.

Sensitivity and specificity are calculated as follows:

$$Sensitivity = \frac{True\ Positives}{True\ Positives + False\ Negatives}$$

$$Specificity = \frac{True\ Negatives}{True\ Negatives + False\ Positives}$$

True positives are high risk individuals correctly classified as high risk; false positives are low

risk individuals incorrectly classified as high risk; true negatives are low risk patients correctly classified as low risk; false negatives are high risk individuals incorrectly classified as low risk.

## Results

### The METABRIC and TCGA-BRCA cohorts lend themselves as use cases for the application of GAN-based prognostic classifiers

One of the major challenges of machine learning applied to -omics data and companion medical records is the imbalance between the high amount of variables compared to the limited number of patients available. Even in the case of breast cancer, one of the most frequent and widely studied malignant neoplasms, this limitation is apparent in the two major public transcriptome datasets, namely the MB cohort (n = 1904 patients, m = 18543 transcripts) and the TCGA cohort (n = 1101 patients, m = 20532 transcripts). This imbalance is exacerbated for prognostic analyses that require long-term (10 years) follow-up information and the application of further exclusion criteria (see methods), reducing cohort sizes to n = 1248 and n = 165, respectively (Fig 1A and 1B). Both cohorts behaved notably different, with patients in the MB cohort on average having an overall substantially better prognosis in overall survival and relapse-free, progression-free or disease specific survival (Fig 1C and 1D). This is likely attributable to the MB dataset largely consisting of stage I and stage II patients (89.5% of patients with reported disease stage at diagnosis), whereas stage III and IV patients are more prominent in the TCGA dataset (40.4% of individuals with available disease stage at diagnosis). Despite these differences, the high risk subgroups of both cohorts showed comparable median survival times (MB = 31.9 months [Fig 1E], TCGA = 26.3 months [Fig 1F]). Due to the limited sizes of these cohorts, they lend themselves as suitably challenging use-cases for applying data augmentation for improving prognostication. In particular, we set out to implement a classifier based on a data augmentation network for improved patient stratification in the MB cohort, to subsequently validate robustness and transferability by integrating the independent TCGA cohort.

### A trained GAN discriminator robustly identifies low and high risk breast cancer patients

To tackle the problem of data scarcity, we implemented a GAN architecture to augment transcriptomic data of the MB cohort and tested the performance of a trained discriminator in stratifying breast cancer patients. First, individual patient transcriptome profiles were rescaled and converted into arrays of pixels (Fig 2A i) in order to use these images as an input for the GAN. Independent of these true patient data, the generator created images representing the transcript profiles of synthetic hypothetical patients together with their category (low or high risk) (Fig 2A ii). After being exposed to a fraction of the real transcriptome images and associated categories, its adversary, the discriminator network then tried to distinguish fake from real transcriptome images and determine the risk category of the synthetic samples (Fig 2A iii). Reiterating this training process over 1000 epochs, the generator learned to create realistic synthetic transcriptome images for high and low risk categories, which then could be used to augment the original training data. Associated characteristics of this process (discriminator loss, discriminator class loss, generator loss) are shown in S1 Fig. Using this approach, the discriminator learned to identify features relevant for the risk category definition, aided by the synthetic profiles that implicitly augmented the real training data at each epoch. The trained GAN discriminator (T-GAN-D) network resulting from this process was frozen and subsequently used as a standalone classifier in a transfer learning fashion. The T-GAN-D categorizes

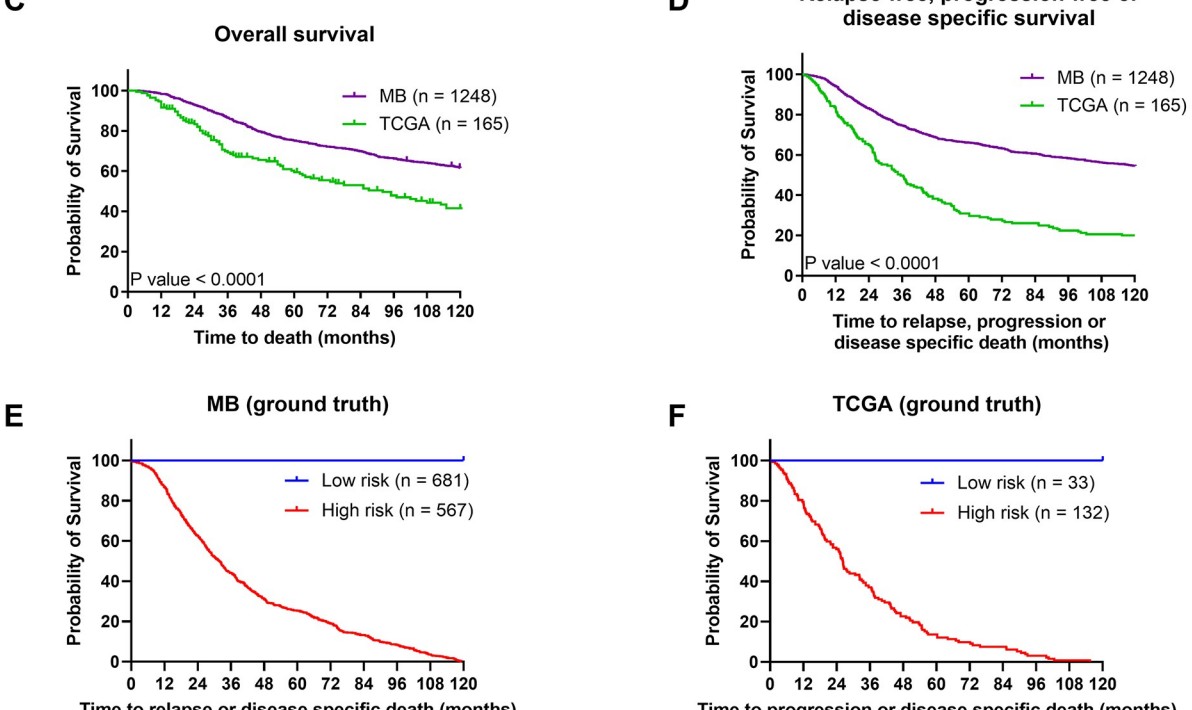

**A** METABRIC sub-cohort (n = 1248)

| Age at diagnosis (years) | # cases | % |
|---|---|---|
| <40 | 83 | 6.7 |
| 40-49 | 188 | 15.1 |
| 50-59 | 288 | 23.1 |
| 60-69 | 392 | 31.4 |
| 70-79 | 235 | 18.8 |
| 80+ | 62 | 5.0 |
| **Disease Stage** | **# cases** | **%** |
| Stage I | 303 | 24.3 |
| Stage II | 514 | 41.2 |
| Stage III | 88 | 7.1 |
| Stage IV | 8 | 0.6 |
| Not available | 335 | 26.8 |
| **Survival (months)** | **Median** | |
| Overall survival | 163.2 | |
| Relapse or disease specific death | 160.2 | |

**B** BRCA-TCGA sub-cohort (n = 165)

| Age at diagnosis (years) | # cases | % |
|---|---|---|
| <40 | 20 | 12.1 |
| 40-49 | 38 | 23.0 |
| 50-59 | 29 | 17.6 |
| 60-69 | 44 | 26.7 |
| 70-79 | 22 | 13.3 |
| 80+ | 12 | 7.3 |
| **Disease Stage** | **# cases** | **%** |
| Stage I | 22 | 13.3 |
| Stage II | 72 | 43.6 |
| Stage III | 50 | 30.3 |
| Stage IV | 14 | 8.5 |
| Not available | 7 | 4.2 |
| **Survival (months)** | **Median** | |
| Overall survival | 92 | |
| Progression or disease specific death | 35.5 | |

**Fig 1. MB and TCGA patient demographics and survival.** (**A**) Patient demographics of the MB subcohort. (**B**) Patient demographics of the TCGA subcohort. (**C**) Overall and (**D**) relapse-free, progression-free or disease specific survival of the MB and TCGA cohorts. (**E**) Kaplan Meier curves comparing low vs high risk patients of the MB and (**F**) the TCGA cohorts.

images from the test fraction of the cohort into the high or low risk categories (Fig 2A iv), thus prognosticating patient outcome. Afterwards, depending on the predicted class, patients were split into high and low risk sub-cohorts which were compared by survival analysis. The generated Log-rank P values were used as the main indicator to assess the stratification performance of the classifier.

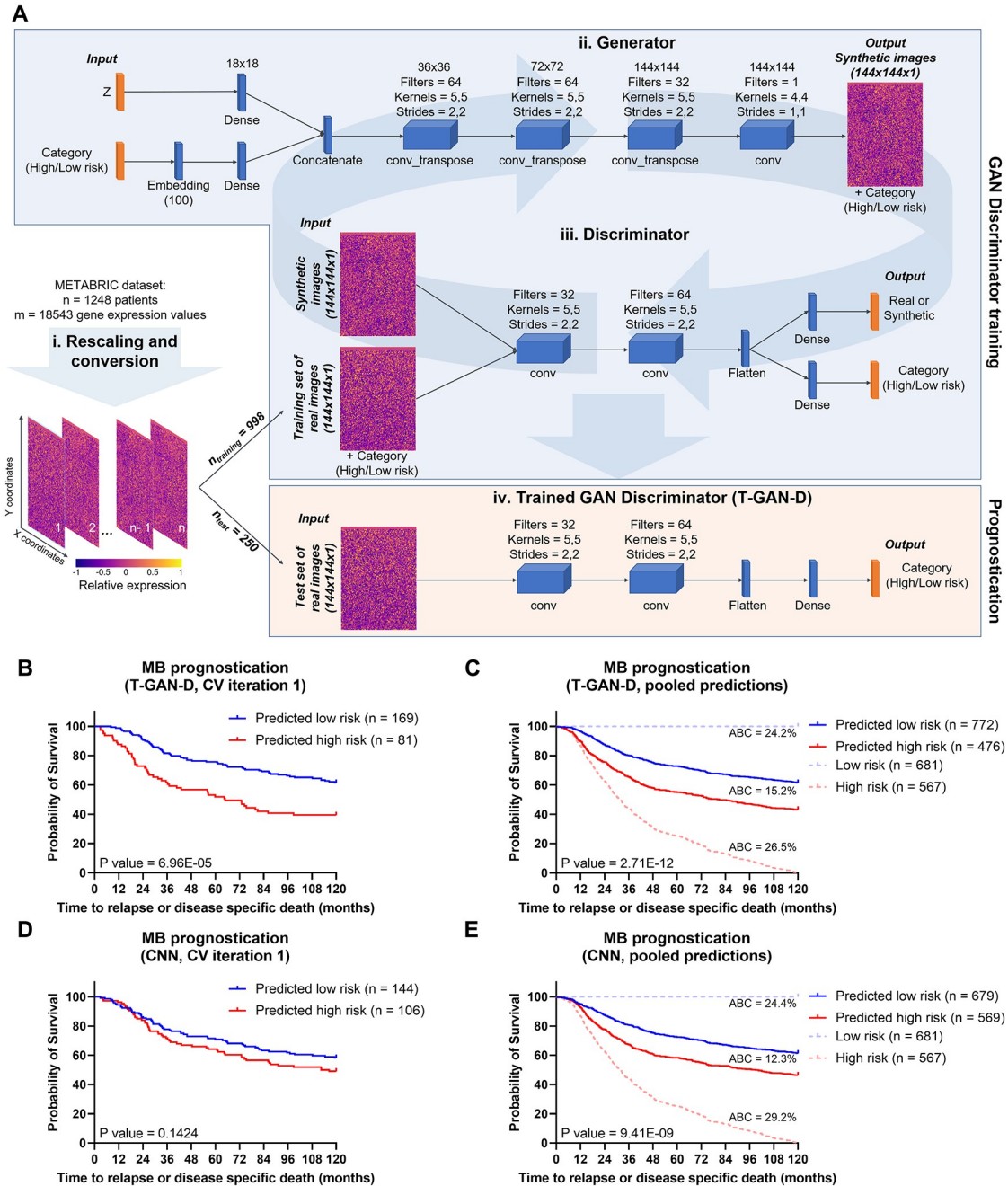

**Fig 2. The T-GAN-D robustly stratifies low and high risk breast cancer patients.** (**A**) Workflow of the data processing, including the schematics of the generator network and its adversary, the discriminator network. Together these result in an AC-WGAN-GP architecture. After the conversion of patient transcriptome profiles into images, 4/5 of the MB dataset was used to train the GAN's discriminator. After 1000 epochs, the trained discriminator was used as a standalone classifier to separate the remaining 1/5 patients of the dataset into low and high risk categories. (**B**) Kaplan-Meier curves separating low vs. high risk patients as predicted with the T-GAN-D (iteration 1 of the 5-fold CV shown as representative). (**C**) Kaplan-Meier curves generated pooling the category predictions obtained for all patients of the MB dataset after five independent CV runs. (**D**) Separation of low vs. high risk patients predicted with a classical CNN on the same subset used in **B** and (**E**) comparison obtained pooling the predictions of five independent CV runs. The area between the curves (ABC) between Low risk (blue dashed line) and Predicted low risk (solid blue line), Predicted low risk and Predicted high risk (solid red line), Predicted high risk and High risk groups (dashed red line) are shown top to bottom in **D** and **E**.

We first implemented and tested the T-GAN-D for its prognostic capability using follow-up and mRNA expression data of the prototyping MB cohort, consisting of n = 1248 individuals and m = 18543 genes. Within this cohort, we independently cross-validated (CV) five-fold with randomly composed training data. Kaplan-Meier curves and log rank testing for each run yielded significant class separations in 4 out of 5 iterations (Figs 2B and S2A). Pooling the results so that each patient of the MB dataset was present once in the survival analysis, the T-GAN-D separated high and low risk patients with high statistical significance (p-value = 2.71E-12) (Fig 2C). To obtain a reference performance baseline, a classical CNN was challenged with the same task, using the same training and test sets for each iteration. The CNN yielded class separations with a p<0.05 in only two out of five iterations (Figs 2D and S2B). In the pooled comparison, the CNN performed well yet failed to outperform the T-GAN-D in separating low vs. high risk patients (Fig 2E and S1 Table). Taken together, these results demonstrate that the reiterative learning process of a GAN to train its discriminator and its use as an independent classifier provides a more robust and slightly improved patient stratification than a classical DL approach.

## Introducing an independent cohort improves MB patient classification

A common limitation of predictors and classifiers is their limited robustness and transferability to independent datasets. This might arise from overfitting or overtraining within the initial cohort but also from heterogeneity and batch effects between source datasets. For validating our approach further, we therefore merged the mRNA expression data of the MB and TCGA cohorts, which originally were quantified with bead-based microarray technology (Illumina Human V3) or RNA-Seq (Illumina HiSeq) platforms respectively [49], by rescaling the expression of transcripts overlapping between the two cohorts (m = 14042). Both the generator and discriminator networks then were retrained using the entire TCGA data plus a fraction of the MB data from the merged dataset and generated predictions on an independent subset of MB patients (Fig 3A), using five-fold cross-validation. The T-GAN-D again separated patients into low and high-risk categories with high statistical significance (Figs 3B and S3A). The CNN tested with the same data performed similarly well, either when trained with real patient data (Figs 3C and S3B) or with an augmented training set containing an additional n × 2 synthetic samples (S4 and S6 Figs). The T-GAN-D trained on the merged and reduced dataset also showed improved accuracy when compared to all settings where both a CNN or the GAN were trained on the full or reduced MB dataset alone (S1–S3 Tables). Therefore, in our setting, rescaling and converting transcriptome profiles into images was sufficient to successfully merge the two cohorts without the need for further preprocessing steps and allowed to stratify patients into high and low risk classes.

## The T-GAN-D outperforms classical outcome predictors and accurately stratifies early stage patients into risk categories

We next compared the performance of CNN and GAN based classifications to other established clinical markers in breast cancer. These included a scoring system based on a multi-transcript signature (Risk-of-recurrence—proliferation, [ROR-P]), estrogen receptor status (ER), human epidermal growth factor receptor 2 status (HER2), and progesterone receptor status (PR). Likewise, tumor staging was included, yet was available for only 911 out of 1248 patients of the MB cohort. The hazard ratios (HR) obtained from a univariate analysis were comparable for ROR-P, HER2 or tumor staging as classifiers, and similar HRs were also obtained for the CNN and T-GAN-D classifiers developed from only the MB transcriptome dataset (Fig 4A). Interestingly, the T-GAN-D classifier resulting from the merged cohort data returned a mean HR>2.0 (+/- 0.4), thereby surpassing all other common markers (Fig 4A) and could compete

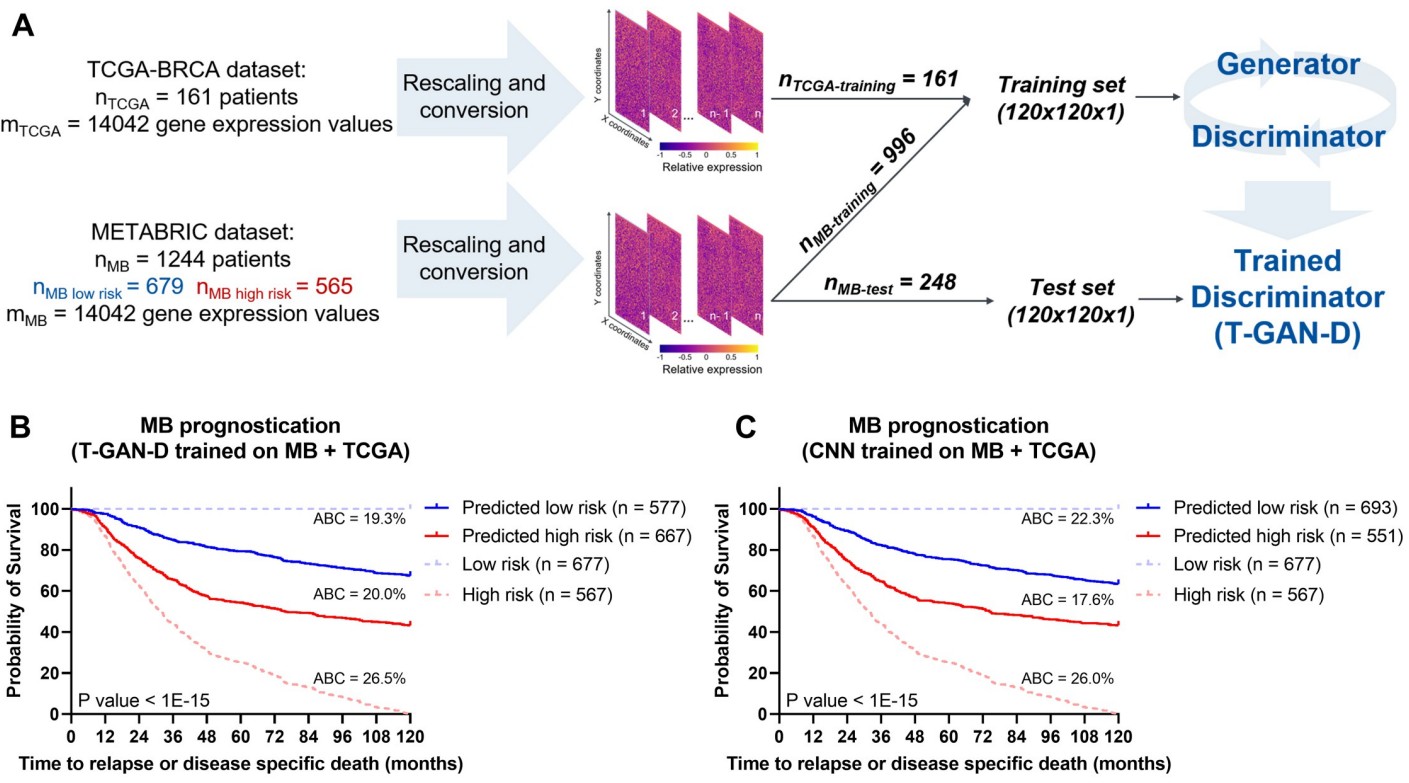

**Fig 3. Introducing the independent TCGA cohort improves MB patient classification.** (**A**) Schematic representing the training strategy: rescaled data from the entire TCGA cohort were merged with 4/5 of the MB cohort to train the T-GAN-D, which was subsequently used to predict the risk class of the remaining 1/5 of MB patients. The process was iterated 5 times. (**B**) Kaplan-Meier curves based on the pooled predictions of the T-GAN-D trained on both cohorts. (**C**) Kaplan-Meier curves separating low vs. high risk patients predicted with the CNN that was trained after merging the MB and the TCGA cohorts. The area between the curves (ABC) between Low risk (blue dashed line) and Predicted low risk (solid blue line), Predicted low risk and Predicted high risk (solid red line), Predicted high risk and high risk groups (dashed red line) are shown top to bottom in **B** and **C**.

with classical ML algorithms (S5 and S6 Figs and S4 Table). This feature was even more pronounced in a multivariate analysis including ER, HER2 and PR biomarkers (Fig 4B). When reducing the MB cohort to those patients for which staging information was available, HRs based on staging and T-GAN-D were comparable (Fig 4C). To test whether both classifiers might be redundant, we performed a T-GAN-D based survival analysis within the tumor stage I and stage II subcohorts, which dominate the MB dataset. T-GAN-D based classification allowed separating high and low risk patients within both tumor stages (Fig 4D and 4E), indicating non-redundancy of the T-GAN-D classification to tumor staging information.

Taken together, these results show enhanced performance of the discriminator network that was exposed to additional synthetic samples during the training process of the GAN. In our use case scenario, the T-GAN-D surpasses individual classical biomarkers and performs well when prognosticating early stage breast cancer cases.

## The T-GAN-D stratifies TCGA patients despite these being scarcely represented

After observing that introducing TCGA patients into the training set of the T-GAN-D did not degrade, but improved the stratification of MB patients, we tested the performance of the classifier on the smaller TCGA dataset. To do this, we retrained the GAN using the entire MB data

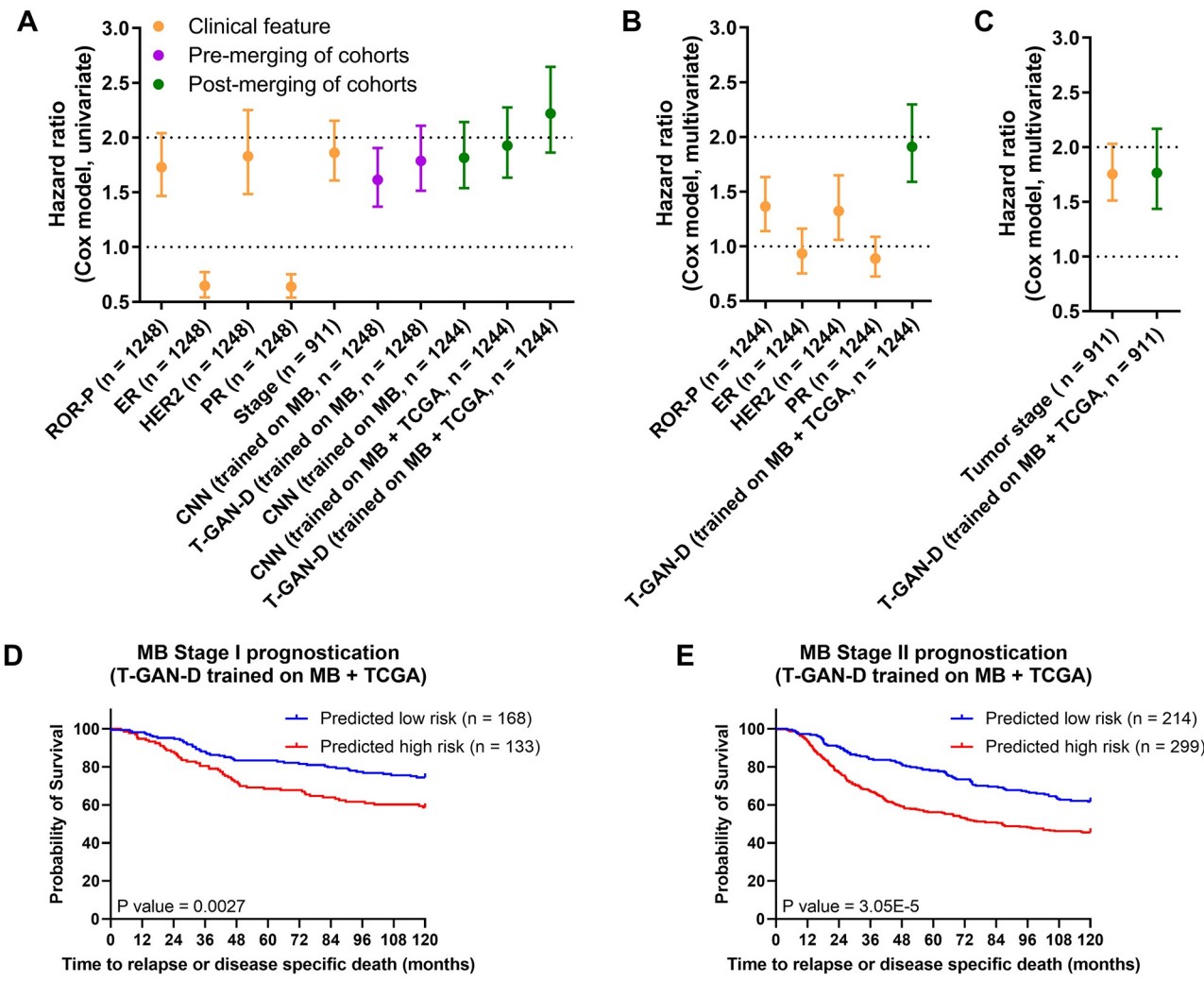

**Fig 4. The T-GAN-D outperforms classical biomarkers after merging the MB and TCGA cohorts and significantly stratifies early stage MB patients.** (**A**) Comparison of the hazard ratios (Cox model, univariate) of a multi-transcript signature (ROR-P) and established prognostic biomarkers (ER, HER2, PR) vs. the CNN and the T-GAN-D before and after cohort merging. (**B**) Multivariate Cox hazard ratio of the T-GAN-D compared to ROR-P and receptor status and (**C**) disease stage. (**D**) Kaplan -Meier curves of Stage I and (**E**) Stage II patients stratified by the T-GAN-D into low and high risk categories.

plus a fraction of the TCGA data from the merged dataset and used the discriminator to generate predictions on an independent subset of TCGA patients (Fig 5A), using five-fold cross-validation. The T-GAN-D correctly predicted 78% of the cases (Figs 5B and S7A and S5 Table). In contrast, when trained on the MB dataset alone, the T-GAN-D was not able to separate high and low risk patients (Figs 5C and S7B), achieving an overall accuracy of only 43% (S5 Table). Therefore, the addition to the training set of a comparably small number of TCGA patients (n = 129) to the larger MB cohort (n = 1244) was sufficient to drastically improve the performance of the T-GAN-D predicting TCGA patient outcome. This demonstrates that even if the training set is largely dominated by patients belonging to one cohort, the introduction of a limited number of samples of a second, differently balanced dataset appears sufficient to possibly capture relevant patterns that contribute to achieving improved prognostic performance.

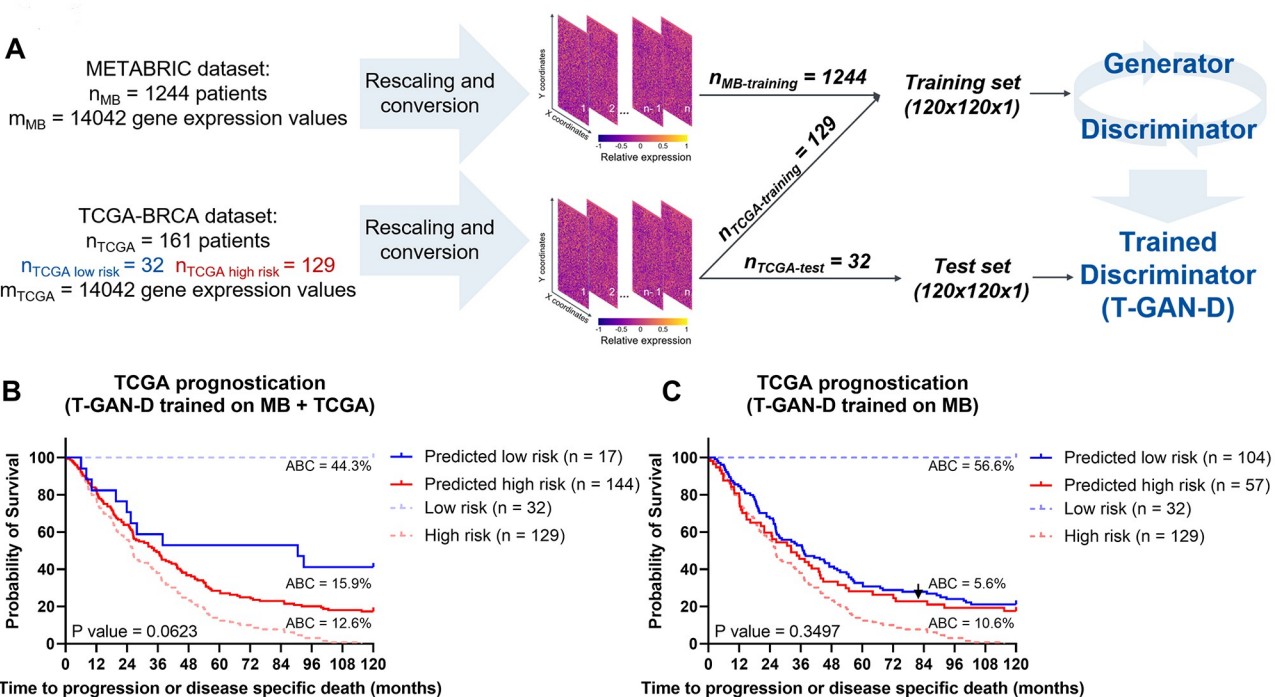

**Fig 5. The T-GAN-D stratifies TCGA patients despite these being scarcely represented in the merged training set.** (**A**) Schematic representing the training strategy: rescaled data from the entire MB cohort were merged with 4/5 of the TCGA cohort to train the T-GAN-D, which was subsequently used to predict the risk class of the remaining 1/5 of TCGA patients. The process was iterated 5 times. (**B**) Stratification of the TCGA patients by T-GAN-D trained on the merged dataset and (**C**) the MB dataset alone. Kaplan-Meier curves were generated pooling the predictions of all iterations of the 5-fold CV. The area between the curves (ABC) between Low risk (blue dashed line) and Predicted low risk (solid blue line), Predicted low risk and Predicted high risk (solid red line), Predicted high risk and High risk groups (dashed red line) are shown top to bottom in **B** and **C**.

## Discussion

The increasing availability and routine acquisition of large scale genomic data encourage the repurposing and application of AI to the field of oncology in order to identify novel means for improved and personalized prediction of prognosis [50]. In this study, we developed a DL-based tool to stratify high vs. low risk breast cancer patients according to full transcriptome profiles. Using the MB and TCGA cohorts as use cases, we converted expression data into images and used the trained discriminator of our GAN architecture as a standalone prognostic classifier. Our results show that the T-GAN-D performed better than classical outcome predictors and maintained robust performance when merging the two cohorts.

AI has already been applied to breast cancer based on different classes of data, to inform diagnosis, treatment planning and prognosis [51,52]. For example, pattern recognition and data augmentation proved to be promising approaches to assist in generating accurate diagnoses from mammography images [53,54]. Transcriptome data were also employed to develop ML-based analysis pipelines for breast cancer subtyping, diagnosis, patient stratification and identification of altered pathways [55], and these techniques may improve the accuracy of cancer prognosis in the future. However, shortcomings must be taken into account, as applicable also to currently available breast cancer datasets. When dealing with low sample size—high dimension datasets such as the MB and TCGA cohorts, common DL classification algorithms such as neural networks may be prone to overfitting [56]. Multi-gene signatures based on the expression of a lower number of transcripts may circumvent this problem, but are applicable only to subsets of patients with specific clinical characteristics [11–14]. To tackle these

problems, we aimed at developing a more universally applicable algorithm that takes advantage of GAN's data augmentation and generalizing capability. In our training strategy, the T-GAN-D was exposed not only to a subset of original data, but also to the synthetic patients generated by the generator in each epoch. This approach for the augmentation of training data was demonstrated before to aid a discriminator in learning hidden features and correlations [57,58]. When compared to a classic CNN, the T-GAN-D showed comparable, yet slightly improved performance. In terms of parameters, the GAN is 2.75 times more complex than the CNN (462,128 vs. 167,490 trainable parameters, respectively), due to the different training process. Nevertheless, trading computational effectiveness with a more stable network can mitigate under- or overfitting problems encountered with CNNs, especially when dealing with small or imbalanced datasets [59].

Other GAN implementations have been applied to the MB or TCGA cohorts in the past, addressing different aims such as the generation of missing data [60], the identification of multi-omics signatures [61] and prognostication [62]. While showing encouraging results, these prior works limited the follow up time to 5 years and focused on death events only. Besides considering longer follow up times, the inclusion of progression or recurrence events in the class definition can be considered a more exhaustive assessment of a patient´s risk category, since OS or DSS alone may be insufficient especially in early stage screenings [63]. In addition, short follow up times were shown to affect the prognostication performance of ML algorithms leading to low sensitivity, mostly due to the insufficient occurrence of recurrence or death events [64].

We demonstrated that the conversion of transcriptome profiles into images allowed the integration of independent transcriptome datasets. To date, the majority of gene expression databases cannot be directly integrated due to different sequencing technologies, protocols or batch effects, with the consequence of producing merely qualitative results in a meta-analysis fashion or unveiling evidences that remain cohort-specific [65]. To test if our conversion strategy could allow a straightforward integration of heterogenous datasets, we challenged the T-GAN-D in assessing the risk category of MB patients, training the network with a subset of MB patients plus the entire TCGA cohort. Introducing patients belonging to a different cohort improved the performance of the classifier, which in our case outperformed established clinical biomarkers and a published ROR-P signature [9] in uni- and multi-variate analyses. The T-GAN-D classifier also stratified early stage breast cancer patients into low and high risk groups, even though no additional factors such as treatment regimens, age, subtype or other clinical features were considered when composing the training datasets. Early stage patients expected to experience recurrence or progression may benefit from more frequent screenings, yet it remains to be assessed if the transcriptome-based classifier operates independently of or correlates with other established risk factors.

High accuracy in predicting the risk class of the smaller and imbalanced TCGA cohort was achieved when training the T-GAN-D with a subset of TCGA patients plus the whole MB dataset. Classical ML algorithms (SVM and RF, among others) were also shown to benefit from the combination of TCGA RNA-Seq and MB microarray data, which in a previous study improved 5 years OS prognostication [66], but lead to misleadingly high accuracy due to highly imbalanced classes. Taken together, our results suggest that the T-GAN-D remains robust when merging cohorts differently balanced between positive and negative outcomes, and that the network is still able to capture relevant risk patterns when one cohort is heavily underrepresented in the training dataset. Therefore, our classification framework may allow the integration of new, smaller datasets, lending itself as a suitable prototype for generating prospective personalized outcome predictions for scarce *de novo* data.

Possible future strategies to improve the performance of our prognostic framework can aim at integrating FS as a preprocessing step. Such methods are commonly employed before the

application of classical ML algorithms to reduce dimensionality, remove redundant or irrelevant variables and improve accuracy [67]. Combining FS with GANs showed increased accuracy compared to FS alone when predicting the vital status of TCGA patients from their transcriptome profiles in a previous study [68]. Similarly, FS was successfully applied before data augmentation to improve the prediction of cancer stage using a CNN [27]. Despite the advantages of reducing the number of variables, the performance of general purpose classifiers such as RF and SVM can vary widely depending on the FS methods employed and the subset of features selected [69], an aspect that can impair the transferability of identified signatures to different cohorts [70]. In addition, discarding features inevitably leads to loss of potentially relevant information, especially when the filtering is not knowledge-driven. To avoid this problem and better exploit the structured nature of images, full transcriptome profiles could be arranged into arrays of pixels organized in a biologically meaningful manner. This could for example be achieved by spatially clustering transcripts of known interaction partners at protein scale. Approaches for data structuring so far included organizing transcripts by gene similarity (computed by t-SNE or kernel PCA) [16], by gene order on chromosomes [71] or by annotated functional hierarchies [72]. The three CNN implementations used in the aforementioned studies performed better than classical ML algorithms when challenged to predict tumor type [16,71] or tumor grade [72] from full transcriptome profiles of the TCGA dataset. Even though no knowledge-driven spatial arrangement of the features was performed in our study, the T-GAN-D performed similarly to general purpose ML classifiers preceded by FS. This suggests that improved performance could possibly be expected when analyzing pixel arrays organized in a biologically meaningful order. Finally, the T-GAN-D lends itself to be tested on multi-omics data retrieved from novel integrative datasets (e.g. mixOmics [73]). To this end, multiple layers of matching -omics data could be converted into separate color channels with identical pixel organization, essentially yielding multichannel images. This approach would allow the analysis of three (or more) separate -omics domains at once, exploiting hidden inter-omics relationships that cannot be captured by classical ML algorithms. Future studies on data augmentation methods and novel architectures (e.g. graph convolutional networks) can be expected to further improve the performance of DL-based classifiers [74].

In conclusion, our proof-of-concept study represents an avenue for developing a scalable data augmentation-based tool that could be a stepping stone towards individualized prognosis in the future. Molecular high throughput techniques are increasing in quality, resolution and amount of data produced and are more and more commonly captured in clinical research and diagnostic environments. It was estimated that within the next decade, between 2 and 40 exabytes of genomic data will be generated every year [75], with large quantities being related to human health and disease. GAN-based approaches therefore could become a meaningful approach to exploit such data for the benefit of patients. In addition, -omics domains other than transcriptomics likewise have the potential to enter the clinical arena as part of routine analytical practice, including proteome, metabolome or lipidome data. Such data classes can readily be integrated with clinical-pathological information [76], and could be processed with the assistance of GAN based approaches to improve patient-tailored interventions or prognostication.

## Supporting information

**S1 Fig. AC-WGAN-GP loss functions. (A)** Loss functions of the discriminator identifying real vs fake patients and **(B)** risk category. **(C)** Loss function of the generator. Loss functions were computed over 1000 training epochs.
(TIF)

**S2 Fig. Kaplan-Meier curves generated with the risk categories predicted in the CV iterations not shown in Fig 2.** The prototyping MB cohort with all available transcriptomic data was used to compare the patient stratification obtained with the (**A**) T-GAN-D and (**B**) a classic CNN.
(TIF)

**S3 Fig. Kaplan-Meier curves of individual CV iterations pooled in Fig 3.** A fraction of the MB and the full TCGA cohorts were integrated to train (**A**) the T-GAN-D and (**B**) the CNN. After rescaling both datasets and filtering out the genes not available in both cohorts the risk class of the MB patients was predicted.
(TIF)

**S4 Fig. Kaplan-Meier curves of individual CV run generated according to the prediction of a CNN trained with augmented data.** (**A**) Synthetic samples were generated exposing the GAN to n = 4/5 of the MB dataset. The CNN was then trained using the original training set plus n*2 synthetic samples. The remaining 1/5 samples were used as a separate test set for the CNN. The process was iterated five times and the resulting survival curves are shown. (**B**) Following the merging of the MB and the TCGA cohort, the GAN was exposed to 4/5 of the MB dataset and the entire TCGA dataset to produce $n \times 2$ synthetic samples. The augmented training set was used to train a CNN and predict the risk category of the remaining unseen 1/5 MB patients. Survival curves were generated according to the predicted class.
(TIF)

**S5 Fig.** Kaplan-Meier curves generated according to the risk class predicted by (A) RF and (B) SVM at each of the 5 CV iteration. Both models were trained with data from the merged MB + TCGA cohorts to predict the risk category of MB patients. At each iteration, a feature selection preprocessing step was performed on the training set. The following number of features was selected at each iteration: $CV_{iteration\ 1} = 32$; $CV_{iteration\ 2} = 33$; $CV_{iteration\ 3} = 33$; $CV_{iteration\ 4} = 40$; $CV_{iteration\ 5} = 40$.
(TIF)

**S6 Fig.** (**A**) Comparison of the hazard ratios (Cox model, univariate) of the CNN trained with augmented data before and after cohort merging and of RF and SVM trained after cohort merging. The dashed line represents the hazard ratio of the T-GAN-D trained after cohort merging. (**B**) Kaplan-Meier curves based on the pooled predictions of the CNN trained with real and synthetic samples before and (**C**) after merging of the cohorts. (**D**) Kaplan-Meier curves based on the pooled predictions of RF and (**E**) SVM trained after merging of the cohorts. The area between the curves (ABC) between Low risk (blue dashed line) and Predicted low risk (solid blue line), Predicted low risk and Predicted high risk (solid red line), Predicted high risk and high risk groups (dashed red line) are shown top to bottom in **B** and **C**.
(TIF)

**S7 Fig. Kaplan-Meier curves of individual CV iterations pooled in Fig 5.** The T-GAN-D was trained (**A**) on the merged dataset and (**B**) on the MB dataset alone. After rescaling both datasets and filtering out the genes not available in both cohorts the risk class of the TCGA patients was predicted.
(TIF)

**S1 Table. Accuracy, sensitivity, specificity and Log-rank P value of each CV iteration and pooled category predictions for the experimental settings displayed in Figs 2 and S2.**
(XLSX)

**S2 Table. Accuracy, sensitivity, specificity and Log-rank P value of each CV iteration and pooled category predictions for the experimental settings displayed in Figs 3 and S3.** (XLSX)

**S3 Table. Accuracy, sensitivity, specificity and Log-rank P value of each CV iteration and pooled category predictions for the experimental settings displayed in S4, S6B, S6C Figs.** (XLSX)

**S4 Table. Accuracy, sensitivity, specificity and Log-rank P value of each CV iteration and pooled category predictions for the experimental settings displayed in S5, S6D, S6E Figs.** (XLSX)

**S5 Table. Accuracy, sensitivity, specificity and Log-rank P value of each CV iteration and pooled category predictions for the experimental settings displayed in Figs 5 and S7.** (XLSX)

## Acknowledgments

**MR and CG** acknowledge the support of the Stuttgart Center for Simulation Science (SimTech).

## Author Contributions

**Conceptualization:** Cristiano Guttà, Christoph Morhard, Markus Rehm.

**Data curation:** Cristiano Guttà.

**Formal analysis:** Cristiano Guttà.

**Funding acquisition:** Markus Rehm.

**Methodology:** Christoph Morhard.

**Resources:** Christoph Morhard, Markus Rehm.

**Software:** Christoph Morhard.

**Validation:** Cristiano Guttà.

**Visualization:** Cristiano Guttà.

**Writing – original draft:** Cristiano Guttà, Markus Rehm.

**Writing – review & editing:** Cristiano Guttà, Markus Rehm.

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
