## [Decision Letter · Decision Letter 0]

23 Nov 2022

Dear Dr. Rehm,

Thank you very much for submitting your manuscript "Applying GAN-based data augmentation to improve transcriptome-based prognostication in breast cancer" for consideration at PLOS Computational Biology.

As with all papers reviewed by the journal, your manuscript was reviewed by members of the editorial board and by several independent reviewers. In light of the reviews (below this email), we would like to invite the resubmission of a significantly-revised version that takes into account the reviewers' comments.

We cannot make any decision about publication until we have seen the revised manuscript and your response to the reviewers' comments. Your revised manuscript is also likely to be sent to reviewers for further evaluation.

Sincerely,

Zhaolei Zhang

Academic Editor

PLOS Computational Biology

Jian Ma

Section Editor

PLOS Computational Biology

Reviewer's Responses to Questions

**Comments to the Authors:**

**Reviewer #1:** This is an interesting study in which the authors develop a predictive algorithm (T-GAN-D) that serves both as a classification and as a data augmentation method, with the goal of improving the stratification of low and high risk breast cancer patients from transcriptomic data. The proposed method is evaluated on two public gene expression datasets, namely the METABRIC and TCGA cohorts. Additionally, the predictive capabilities of the T-GAN-D are compared with the performance achieved by a CNN and several established breast cancer biomarkers. Finally, the work is reproducible, since all the material needed to replicate the results obtained in this study is publicly available to the scientific community.

However, some important concerns must be addressed to meet the publication criteria:

1. The rationale of converting gene expression data to arrays of numeric values needs to be further justified. In contrast with an image, a gene expression sample has an unstructured nature and lacks any local information that can be exploited by convolutional layers. In this way, authors should not only compare the performance of the T-GAN-D with a CNN, but also with the performance obtained by traditional ML models (such as SVM, random forest, etc.) that use gene expression vectors (without any image transformation) as input data. To counteract the great imbalance between the number of variables (high) and the number of samples (low), authors could use traditional feature selection methods to reduce the number of variables used as input to the classical ML classifiers.

2. Accuracy is used as a metric to evaluate the classification performance of T-GAN-D and CNN. However, additional metrics, e.g., specificity, sensitivity, should also be computed to perform a thorough evaluation of the classification capabilities of the models, by evaluating their performance separately for each of the two classes (high and low risk). This is specially relevant when evaluating the models on the TCGA dataset (last subsection of Results section), where there is a considerable difference between the number of high and low risk patients.

Minor comments:

- Line 243: “and” should be changed for “an”.

- Fig. 1F: “Predicted” should be removed from the legend of the figure.

- In Fig. 4A, authors need to clarify the meaning of “Pre” and “Post” transcripts filtering.

**Reviewer #2: **This manuscript proposed to use of a GAN-based data augmentation strategy to alleviate the overfitting of the classification model, with the aim of predicting the prognostic risk of breast cancer patients. The manuscript is in general well-written but needs to be checked thoroughly. I may have some concerns that need to be addressed.

Can you provide the number of patients and genes in both the MB cohort and TCGA cohort before and after defining the low and high-risk categories?

Please explain the rationality of the category definition of high-risk and low-risk patients.

How to convert the expression values of 18543 genes into the matrix of size 144*144?

What is the difference in time complexity between T-GAN-D and CNN?

It shows 161 patients are predicted low-risk group and high-risk group in both Fig.5 (B and C). I think the experiments resulted from the TCGA-BRCA dataset instead of MB + TCGA, or MB.

Current methods under comparison do not provide a conclusive benchmark.

Many descriptions are not standardized. such as inconsistent words (BRCA-TCGA and TCGA-BRCA).

**Reviewer #3: **The paper shows the application of a generative adversarial network (GAN) to improve prognosis in breast cancer using deep learning (DL). Specifically, the authors use transcriptomic data to identify low-risk and high-risk breast cancer patients. They process the gene expression data to turn it into images that can be more easily used by two DL models: a convolutional network (CNN) and an Auxiliary Classifier Wassertein GAN with gradient penalty (AC-WGAN-GP). The METABRIC and TCGA-BRCA selected cohorts are two of the largest and best-annotated breast cancer databases, an excellent choice for testing the efficacy of models. The paper shows that the discriminator of the AC-WGAN-GP model trained with the real processed images and the synthetic images created by the generator, as is logical in the normal process of a GAN, can be used as a DL classifier model to identify the patient prognosis. This discriminator, named by the authors as T-WGAN-D, improves the results obtained by a CNN. It should be noted that the paper does not present a new model or data augmentation (DA) application, but rather an application variant of part of an AC-WGAN model (the discriminator). The paper uses a non-precise definition of DA. It is not correct to consider DA the use of synthetic images in the training of a model that precisely requires these for its training. If the generated synthetic images were also used to augment the training set of another model, such as the CNN, it could be considered DA. Beyond this critique, the application of T-WGAN-D is novel enough considering the problem posed and it improves the performance achieved by the CNN model.

I pointed out some issues that needs to be addressed:

- A paragraph that explains structure of paper can be added at the end of the introduction section.

- Other references that also studied GAN-based data augmentation techniques on TCGA mRNA data can be added.

o Moreno-Barea, F.J., Jerez, J.M., Franco, L. (2022). GAN-Based Data Augmentation for Prediction Improvement Using Gene Expression Data in Cancer. In International Conference of Computational Science – ICCS 2022. Lecture Notes in Computer Science, vol 13352. Springer, Cham. doi: 10.1007/978-3-031-08757-8_3

o Kwon C, Park S, Ko S, Ahn J. Increasing prediction accuracy of pathogenic staging by sample augmentation with a GAN. PLoS One. 2021 Apr 27;16(4):e0250458. doi: 10.1371/journal.pone.0250458

- The authors should compare the performance of T-WGAN-D against other traditional methods and models used to identify high-risk breast cancer patients.

- The authors should try data augmentation on CNN, as increased variety may lead to improved performance. DA involves using synthetic data to train models that do not use it primarily.

- At line 243, the authors wrote " Introducing and independent cohort improves MB patient classification." Please review this error.

- In Figure 4.A it appears that AC-GAN was used, but T-GAN-D is mentioned in the text. Please review it.

- In the section on the results of the union of both cohorts to predict on one of them, the re-training of only the discriminator is mentioned. However, in Figures 3 and 5 the generator training also appears. These create confusion please revise accordingly.

- It would be convenient to mention transfer learning in the section described in the previous point. The retraining procedure is transfer learning, not DA.

- Future studies can be added with possibility of employing other deep learning architectures.

**Have the authors made all data and (if applicable) computational code underlying the findings in their manuscript fully available?**

Reviewer #1: Yes

Reviewer #2: Yes

Reviewer #3: Yes

PLOS authors have the option to publish the peer review history of their article (what does this mean?). If published, this will include your full peer review and any attached files.

Reviewer #1: No

Reviewer #2: No

Reviewer #3: No
---

## [Decision Letter · Decision Letter 1]

4 Mar 2023

Dear Dr. Rehm,

Thank you very much for submitting your manuscript "Applying a GAN-based classifier to improve transcriptome-based prognostication in breast cancer" for consideration at PLOS Computational Biology. As with all papers reviewed by the journal, your manuscript was reviewed by members of the editorial board and by several independent reviewers. The reviewers appreciated the attention to an important topic. Based on the reviews, we are likely to accept this manuscript for publication, providing that you modify the manuscript according to the review recommendations.

Sincerely,

Zhaolei Zhang

Academic Editor

PLOS Computational Biology

Jian Ma

Section Editor

PLOS Computational Biology

Reviewer's Responses to Questions

**Comments to the Authors:**

Reviewer #1: Overall, the authors have produced an improved revised version of the manuscript. However, some concerns must be addressed to meet the publication criteria:

1. If we compare the results obtained by the traditional machine learning (ML) models (Supplementary Figures 5 and 6, as well as Supplementary Table 4) and the proposed T-GAN-D model, the T-GAN-D approach does not outperform the support vector machine (SVM) algorithm. Consequently, authors should justify in the manuscript the development of a complex deep learning (DL)-based methodology when a simpler and straightforward approach based on a feature selection method and a classical ML algorithm achieves an equivalent level of performance.

2. Authors should describe in the manuscript how sensitivity and specificity metrics were calculated, i.e. in the context of binary classification, which category (either high or low risk) was considered as positive, and which one was considered as negative.

Reviewer #2: The authors have answered my comments clearly.

Reviewer #3: The article shows many improvements compared to the first revised version. All the issues raised by the reviewers have been addressed and the scientific and technical quality of the article has increased with the changes introduced.

**Have the authors made all data and (if applicable) computational code underlying the findings in their manuscript fully available?**

Reviewer #1: Yes

Reviewer #2: Yes

Reviewer #3: Yes

PLOS authors have the option to publish the peer review history of their article (what does this mean?). If published, this will include your full peer review and any attached files.

Reviewer #1: No

Reviewer #2: No

Reviewer #3: No

Figure Files:

Data Requirements:

Reproducibility:

References:

---

## [Editor Report · Decision Letter 2]

17 Mar 2023

Dear Dr. Rehm,

We are pleased to inform you that your manuscript 'Applying a GAN-based classifier to improve transcriptome-based prognostication in breast cancer' has been provisionally accepted for publication in PLOS Computational Biology.

Best regards,

Zhaolei Zhang

Academic Editor

PLOS Computational Biology

Jian Ma

Section Editor

PLOS Computational Biology

---

## [Editor Report · Acceptance letter]

28 Mar 2023

PCOMPBIOL-D-22-01501R2 

Applying a GAN-based classifier to improve transcriptome-based prognostication in breast cancer

Dear Dr Rehm,

I am pleased to inform you that your manuscript has been formally accepted for publication in PLOS Computational Biology. Your manuscript is now with our production department and you will be notified of the publication date in due course.

With kind regards,

Anita Estes
